# Lowering of the singlet-triplet energy gap via intramolecular exciton-exciton coupling

Clara Schäfer [1,5], Rasmus Ringström [2,5], Jörg Hanrieder [3,4], Martin Rahm [2], Bo Albinsson [2] & Karl Börjesson [1] ✉

Organic dyes typically have electronically excited states of both singlet and triplet multiplicity. Controlling the energy difference between these states is a key factor for making efficient organic light emitting diodes and triplet sensitizers, which fulfill essential functions in chemistry, physics, and medicine. Here, we propose a strategy to shift the singlet excited state of a known sensitizer to lower energies without shifting the energy of the triplet state, thus without compromising the ability of the sensitizer to do work. We covalently connect two to four sensitizers in such a way that their transition dipole moments are aligned in a head-to-tail fashion, but, through steric encumbrance, the delocalization is minimized between each moiety. Exciton coupling between the singlet excited states considerably lowers the first excited singlet state energy. However, the energy of the lowest triplet excited state is unperturbed because the exciton coupling strength depends on the magnitude of the transition dipole moments, which for triplets are very small. We expect that the presented strategy of designed intramolecular exciton coupling will be a useful concept in the design of both photosensitizers and emitters for organic light emitting diodes as both benefits from a small singlet-triplet energy gap.

Triplet photosensitizers (PS) are crucial for photocatalysis[1–4], triplet-triplet annihilation upconversion[5–7], photodynamic therapy[8–11], and various other fields and applications. New avenues of utilizing PS are frequently discovered and with the growing interest, the development of organic, highly efficient PS has gained more attention. The crucial property of a triplet PS is its rate of intersystem crossing (ISC). ISC is a spin forbidden process which can be realized by magnetic perturbations, like spin-orbit coupling[12]. To achieve a high rate of ISC, traditional PS make use of the heavy atom effect, known to enhance spin-orbit coupling[13,14]. Heavy elements like Pt, Pd, Ru, Ir, Se, I and Br have shown to enhance ISC[15,16]. However, there are considerable disadvantages of using heavy atoms to increase the rate of ISC, such as their scarceness, toxicity, and cost. Furthermore, the increased spin-

orbit coupling also reduces the triplet state lifetime, which is a negative feature when used in PS applications[17]. To avoid some of these drawbacks, development and improvement of organic, heavy atom free triplet PS is in high demand.

Several strategies have been developed to realize an enhanced rate of ISC for compounds without incorporating heavy atoms. One that has gained recognition is spin-orbit charge transfer intersystem crossing (SOCT-ISC)[9,18,19]. Systems exhibiting this phenomenon are usually donor-acceptor dyads able to perform intramolecular excited state electron transfer reactions. Upon photoinduced electron transfer, the system forms a charge separated (CS) state. The donor and acceptor moieties are designed to have an orthogonal geometry towards each other. Due to the orthogonality of the system, the change

[1]Department of Chemistry and Molecular Biology, University of Gothenburg, Box 462, 405 30 Gothenburg, Sweden. [2]Department of Chemistry and Chemical Engineering, Chalmers University of Technology, Kemivägen 10, 412 96 Gothenburg, Sweden. [3]Department of Psychiatry and Neurochemistry, Institute of Neuroscience and Physiology, Sahlgrenska Academy at the University of Gothenburg, Mölndal Hospital, House V3, 431 80 Mölndal, Sweden. [4]Department of Neurodegenerative Disease, Queen Square Institute of Neurology, University College London, London, WC1N 3BG London, UK. [5]These authors contributed equally: Clara Schäfer, Rasmus Ringström. ✉e-mail: karl.borjesson@gu.se

in molecular orbital angular momentum compensates for the change in spin angular momentum, enhancing the rate of ISC[20]. Thus, from the CS state the molecule can undergo charge recombination to the triplet state in an efficient manner. This approach does not rely on the heavy atom effect and is therefore highly appropriate for the design of heavy atom free photosensitizers.

Recently, several triplet photosensitizers undergoing SOCT-ISC based on the BODIPY scaffold have been reported[5,8,21–23]. Within the scope of these reports, a BODIPY-anthracene dyad stands out with a high yield of ISC[24–26]. These studies also highlight the importance of the energy alignment between the $S_1$ and CS states. The energy of the CS state is strongly dependent on the polarity of the solvent, and the $S_1$-CS energy alignment can therefore be tuned by the choice of solvent. In a non-polar solvent (toluene), the CS state was at a higher energy than the $S_1$ state, resulting in a neglectable ISC but a high yield of fluorescence ($\Phi_{Fluo} = 0.81$). However, with increasing solvent polarity, the CS state is stabilized, resulting in exergonic excited state electron transfer. The yield of ISC thus increased ($\Phi_{ISC} = 0.92$) at the expense of the fluorescence ($\Phi_{Fluo} = 0.002$)[24].

Organic dyes such as BODIPY's have a relatively large $S_1$-$T_1$ energy gap. In all applications relying on a triplet sensitizer, this gap can be regarded as an energy loss. As small an energy gap as possible is thus preferable. In recent years, the field of organic electronics has been focusing on reducing this energy gap by separating the HOMO and LUMO orbitals in space as doing so affects the electron exchange interaction[12,27–29]. However, there are other possible mechanisms that theoretically can perturb the $S_1$-$T_1$ energy alignment[30,31]. One of these is exciton coupling, the effect seen in J-aggregates[32]. Exciton coupling influences the singlet state energy without affecting either the triplet or CS state energies significantly. Even before the development of the theory of exciton coupling it was observed that aggregation affects the fluorescence and phosphorescence differently[33]. Curiosity about this observation amongst other things, later lead to the finding that exciton coupling influences the singlet state energy without affecting either the triplet or CS state energies significantly[34]. This is because the effect is based on a Columbic coupling of the transition dipole moments associated with the respective states, and only singlet states have an appreciable magnitude of their transition dipole moments[35]. Thus, from a theoretical perspective, assembling an organic PS in the form of a J-aggregate, will reduce energy losses by allowing a red-shifted excitation wavelength.

Here, BODIPY-anthracene dyads were β-tethered into oligomers (2–4 units) in a non-conjugated fashion. The β-tethering aligns the $S_0$-$S_1$ transition dipole moments of the BODIPYs in a 'head-to-tail' fashion. Intramolecular exciton coupling thus occurs, giving an allowed low energy transition that depends on the number of coupled BODIPY-anthracene moieties. This is the same effect as seen in J-aggregates and has also previously been observed in ethylene-bridged BODIPY oligomers[36,37]. The β-tethered oligomers have almost unperturbed energies of their $T_1$ and CS states because direct electronic communication between the moieties was designed to be suppressed. However, they are still able to perform ISC. The formation of covalently tethered J-aggregate mimics is thus a viable path to reduce energy losses in organic sensitizers. This by selectively lowering the $S_1$ state without significantly influencing the energy of the triplet state.

## Results

To lower the $S_1$ energy without affecting the CS and $T_1$ energies, it is important to align the $S_0$-$S_1$ transition dipole moments without causing an extension of the aromatic system in oligomers. This can be achieved if steric hindrance forces the individual moieties into an orthogonal geometry. The BODIPY anthracene dyad was synthesized following the literature procedure[38]. Dimerization of BODIPY dyes has previously been achieved in the meso, α, and β positions[39–41]. Of these attachment

positions, β-tethering also leads to oligomerization of BODIPY dyes[42–44]. Both conjugated oligomers[42] as well as non-conjugated oligomers, where methyl groups on adjacent carbons result in a large dihedral angle between moieties due to steric hindrance[43,44], have been accessed. The conjugated BODIPY oligomers were obtained via Suzuki-Miyaura coupling. The benefit of this reaction is the controllability of the degree of oligomerization. However, a fully conjugated system would not just influence the singlet energy, but also the triplet energy to a similar extent. The non-conjugated systems were obtained by treating the monomer with either the hypervalent iodine reagent phenyliodine(III)bistrifluoroacetate (PIFA), or anhydrous iron(III) trichloride (Fig. 1). In both cases, trimers were the highest order of oligomers isolated[43,44]. However, polymerization was achieved using FeCl₃, when leaving the reaction for a longer time. Both conditions were tested here, resulting in the dimer regardless of conditions used (Supplementary Note 1, 2, Supplementary Figs. 1–19, Supplementary Tables 1, 2). The reaction using FeCl₃ yielded the dimer to a larger extent, 17% as compared to 8% when using PIFA. Furthermore, the reaction using FeCl₃ also yielded higher oligomers, the trimer, and the tetramer. Atropisomers of the trimer could be isolated. These are hypothesized to correspond to the cis and trans isomers with regards to the anthracene units on the end BODIPYs and show discernible ¹H NMR spectra (Supplementary Figs. 11, 14; unfortunately, the distance in between the anthracene units is too large to obtain NOESY signals, which would allow for identification of the two different isomers). After purification, only trace amounts, 1–2% of the two atropisomers of the trimer, and >1% of the tetramer were obtained. However, the small amounts collected were enough to confirm their structure through ¹H, ¹³C (Supplementary Dataset 1), diffusion NMR and MALDI, and to probe the excited state energetics and dynamics.

### The energies of the singlet and triplet states

In the presentation of the excited state energetics, we start by discussing the $S_1$ state. We then continue with the $T_1$ and the CS states, so that we in the conclusion can discuss the complete energetic picture of our systems. To obtain information on the $S_1$ state energy, and to see if it could be decreased successfully, the absorption spectra of the oligomers were analyzed. The two conformers of the trimer exhibit identical photophysical properties. In this discussion, the results are therefore combined as the trimer. The absorption envelope (Fig. 2a) of all compounds can be deconvoluted into an anthracene part (370–400 nm) and a BODIPY part (500–600 nm). The wavelength (i.e., energy) of the anthracene absorption is not affected by

**Fig. 1 | The BODIPY-anthracene dyad and its oligomers.** Synthesis of the meso-10-phenylanthracene-BODIPY dimer (n = 1), trimer (n = 2) and tetramer (n = 3). Reaction conditions: (Iₐ) PIFA, BF₃·OEt₂, DCM, −78 °C · r.t, 1 h. (I_b) anhydrous FeCl₃, DCM, r.t., 25 min. The dimer was obtained using Iₐ and I_b. The trimer and tetramer were obtained using I_b. The $S_0$-$S_1$ transition dipole moments for the anthracene and BODIPY units are depicted as red, dashed arrows[24].

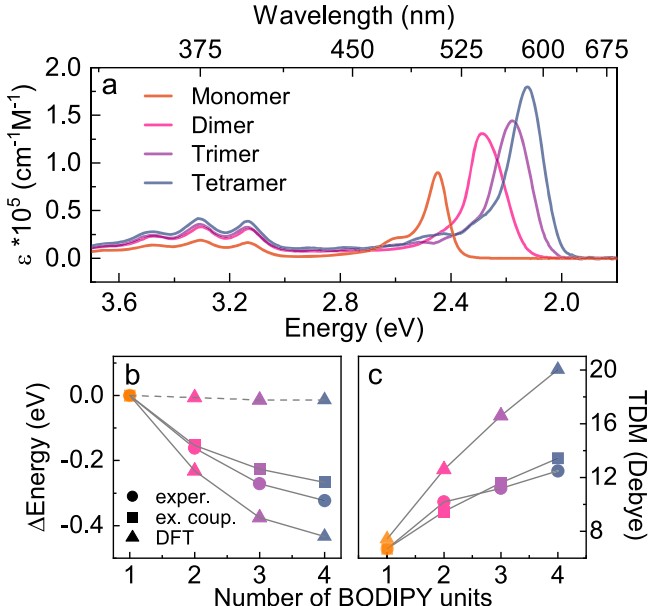

**Fig. 2 | Energetics of excited states. a** Molar absorptivity spectra of the synthesized oligomer series recorded in DCM. **b** The relative energy of the absorption maximum versus number of BODIPY units in the oligomers. Experimental values (circles) are compared to theoretical values based on a point dipole interaction (squares) and DFT (triangles, solid line for $S_1$ and dashed line for $T_1$). **c** Transition dipole moments (TDMs) versus number of BODIPY units in the oligomers. Experimental values (circles) are compared to DFT calculations (triangles), and the exciton coupling model (squares) where the magnitude of the monomer transition dipole moment is scaled with the square root of the number of monomer units. Source data are provided as a Source Data file.

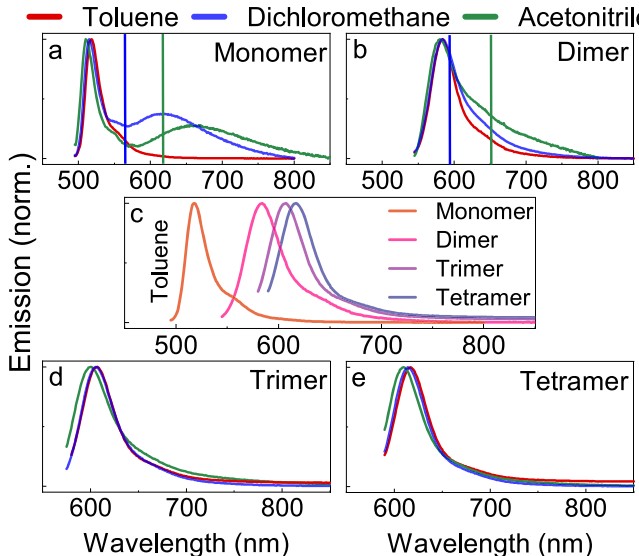

**Fig. 3 | Emission from monomer and oligomers.** Emission of the monomer (**a**), dimer (**b**), trimer (**d**) and tetramer (**e**) are shown in toluene, DCM and ACN. Emission of all oligomers in toluene are shown in (**c**). The vertical lines show the energy of the CS state in DCM (blue) and ACN (green) estimated from electrochemistry. Source data are provided as a Source Data file.

oligomerization. This means that the anthracene moieties are located far enough apart, and/or the orientation between them is disadvantageous, for Columbic interactions between the transition dipole moments to become significant. The situation for the BODIPY part, where the transition dipole moments are designed to be in a head-to-tail fashion, is quite different. The absorption of the oligomers is shifted to longer wavelengths, with absorption maxima at 506, 542, 569, and 583 nm for the monomer, dimer, trimer, and tetramer, respectively. As expected, the absorption energy converges as the number of moieties increases. The excitation energy thus decreases asymptotically, reducing the gain in making even longer oligomers (Fig. 2b). Furthermore, the magnitude of the transition dipole moments were calculated from the molar absorptivity spectra (Supplementary Note 3, Supplementary Fig. 20)[45]. As expected for an exciton coupled system, the transition dipole moments scale well with the square root of the number of moieties (Fig. 2c)[35], although limited purity of the tetramer (Supplementary Fig. 17) is exhibited as a less than expected transition dipole moment for that molecule.

Previously reported conjugated BODIPY dimers and trimers, show a different trend than we observe here. Much larger bathochromic shifts have been reported for conjugated oligomers[42]. The different observations indicate different main communication pathways for conjugated and non-conjugated oligomers. In the latter, the lack of conjugation in between the monomer units minimizes electron delocalization. Exciton coupling could however explain the observed spectral changes, as it does not involve conjugation between moieties[46]. Using a point dipole approximation, the Coulombic interaction energies between moieties in the oligomers were calculated (Supplementary Note 4). Based on this, the spectral shift solely due to Coulombic interactions could be extracted. The vast majority of the experimentally observed spectral shift can be reproduced by taking Coulombic interactions into account. Thus, the observed spectral

changes are mainly a consequence of exciton coupling rather than electronic delocalization.

Quantum mechanical calculations were performed to validate the degree to which the triplet energy remains constant across the series. The geometry of singlet and triplet ground states were optimized in vacuum at the $\omega$B97XD/6-31 g(d) level of theory (Supplementary Fig. 21, Supplementary Dataset 2) and singlet excitation energies were obtained through TD-DFT using the singlet ground state geometry using Gaussian 16, revision B.01[47,48]. No imaginary frequencies for the optimized ground state geometry verifies that a minimized structure was obtained. Although the overall trend is captured well, these calculations overestimate the reduction of the $S_1$ energy as well as the increase of the transition dipole moment with oligomer length (Fig. 2b, c, Supplementary Table 3). But, more importantly, the calculations predict that the energy of the $T_1$ states only marginally decrease across the monomer-to-tetramer series (Fig. 2b).

### Emission efficiency and dynamics

We will now turn our attention to the energy relaxation pathways towards the charge separated state after exciting the oligomers. Figure 3 shows the steady state emission spectra of the monomer and oligomers in toluene (for other solvents and oligomers, see Supplementary Figs. 22–26). Clear BODIPY like emission envelopes that follow the same energetic trend as the absorption spectra can be seen. However, when increasing the polarity of the solvent, a distinguishable second band is observed. This has, for the monomer, previously been ascribed to emission from the charge separated state[24]. The emission from the CS state is intense and spectrally well resolved for the monomer, but it is also present to a minor degree in the oligomers as a shoulder at lower energies in the high polar solvents (Fig. 3).

It is difficult to draw any conclusions regarding the efficiency of reaching the CS state based on the magnitude of the CS emission. It is better to compare how well the presence of an energetically accessible CS state quenches the fluorescence (Table 1). The fluorescence quantum yield of the monomer shows a drop from 0.80 in toluene to 0.02 in DCM and 0.01 in ACN. This effect can be explained by the polarity and thus the dielectric constants of the solvents. The Rehm-Weller equation predicts that the energy of the CS state lowers as the dielectric constant increases[15,49]. ACN has a larger dielectric constant

**Table 1 | Spectroscopic data of the BODIPY oligomers collected in different solvents**

| Compound | Solvent | ε (M$^{-1}$cm$^{-1}$) at λ$_{Abs}$ max | λ$_{Abs}$ max (nm) | λ$_{Em}$ max (nm) | Φ$_{Em}$ | τ$_{Em}$ (ns) | τ$_{T1}$ (μs) | Φ$_{ISC}$ |
|---|---|---|---|---|---|---|---|---|
| Monomer | Toluene | | 509 | 518 | 0.80[a] | 5.29[c] | - | - |
| | DCM | 88,000 | 506 | 516/622 | 0.05[a,f] | 5.17[c,d]/4.99[c] | 165[g] | 0.9[h] |
| | ACN | | 503 | 512/660 | 0.02[a,f] | 5.90[c,d]/3.11[c] | 142[g] | 0.92 |
| Dimer | Toluene | | 545 | 584 | 0.70[b] | 3.05[e] | - | - |
| | DCM | 130,000 | 542 | 584 | 0.54[b] | 3.68[e] | 418 | 0.17 |
| | ACN | | 537 | 580 | 0.01[b] | 0.51[e] | 447 | 0.14 |
| Trimer | Toluene | | 573 | 607 | 0.72[b] | 2.29[e] | - | - |
| | DCM | 145,000 | 569 | 606 | 0.62[b] | 2.65[e] | 425 | 0.1 |
| | ACN | | 563 | 600 | 0.05[b] | 0.57[e] | - | - |
| Tetramer | Toluene | | 586 | 617 | 0.55[b] | 1.90[e] | - | - |
| | DCM | 180,000 | 583 | 614 | 0.45[b] | 2.09[e] | 105 | 0.07 |
| | ACN | | 576 | 611 | 0.07[b] | 0.66[e] | 100 | 0.08 |

The probing wavelength for emission decays was close to the emission maximum, and for transient absorption the maximum of the ground state bleach. Note that as the S$_1$ and CS states in some cases are connected in a dynamic equilibrium, it is not possible to calculate the radiative rate constant of fluorescence from the yield of fluorescence and the fluorescence lifetime.
[a]Fluorescein in 0.1 M NaOH (Φ$_f$ = 0.91)[57] was used as reference compound for the fluorescence quantum yield determination (excitation at 491 nm, refractive index: 1.33).
[b]The monomer was used as reference compound for the fluorescence quantum yield determination (excitation at 375 nm) (Refractive index: DCM: 1.413; Toluene: 1.497; ACN: 1.344).
[c]Fluorescence lifetime was measured upon excitation at 510 nm.
[d]A second decay was observed which was under-resolved by our instrumentation (Supplementary Fig. 26).
[e]Emission lifetime was measured upon excitation at 375 or 377 nm.
[f]The CS emission was fitted to a Gaussian function (on the energy scale) and subtracted from the total emission to extract the BODIPY fluorescence quantum yield contribution to 0.02 and 0.01 in DCM and ACN, respectively. These values were used in the temperature dependent study and compared to simulations.
[g]Triplet lifetime obtained using Eq. 2 due to triplet-triplet annihilation affecting the observed triplet kinetics (see "Methods" for more information).
[h]The literature value was used[24], and this compound also served as reference in the measurements.

than DCM and toluene. Therefore, the CS state in ACN is lower than in DCM, which is lower than in toluene. The driving force for populating the CS state is thus larger in ACN and DCM, explaining the rise of the second emission band in these solvents. The drop in Φ$_{Em}$ is less intense for the dimer, trimer and tetramer in DCM, but it is still considerable in ACN. From a driving force perspective, this difference indicates that the energy of the singlet state decreases more than the CS state in the oligomers. Thus, the energy difference between the S$_1$ and CS states only remains large enough in ACN for charge separation to out-compete fluorescence in the longer oligomers.

To evaluate the efficiency of internal conversion from higher excited states towards the S$_1$ state, different excitation energies were used to excite the monomer. Excitation of the BODIPY moiety is described above. Upon excitation at higher energies, at the anthracene absorption, the emission spectrum looks almost identical (Supplementary Fig. 22). The only difference is seen in the relative emission intensity between the S$_1$ and the CS states emission bands. When the anthracene unit is excited, the CS state emission is relatively increased. This indicates that charge separation competes with S$_n$ to S$_1$ internal conversion in these systems.

The fluorescence quantum yield of the monomer is greatly reduced in ACN compared to toluene. However, the observed emission lifetime is not reduced to the same extent (Table 1, see Supplementary Figs. 27–33 for emission decays). This surprising feature has been previously observed[24], and we will try to rationalize it. When recording the emission lifetime for the monomer, a long-lived transient as well as an instrument response function limited signal is present when monitoring the S$_1$ but not the CS state emission (Supplementary Figs. 27, 29). The rate of charge separation is thus faster than the time resolution of our instrumentation. The long-lived emission must therefore not be due to the directly excited BODIPY unit. Furthermore, when observing the steady state emission as a function of decreasing temperature (Supplementary Fig. 34), we see a reduction of the S$_1$ emission until a plateau value is reached. Based on these two observations, we suggest that the plateau value represent prompt emission from the S$_1$ state, and the temperature dependent emission represents endothermic charge recombination from the CS state back to the S$_1$ state. To strengthen this hypothesis, the system

was modeled using rate equations (Supplementary Note 5). In the modeling, it was assumed that the S$_1$ and CS states are connected by a microscopic reversibility, implying that the ratio of the rate constants between them being equal to the Boltzmann factor. The modeled system thus strongly resembles that used when modeling molecules exhibiting E-type delayed fluorescence (with the triplet state then being replaced by a CS state). Using the energy difference between the two states as a fitting parameter, both the steady state and time resolved temperature dependent emission could be qualitatively reproduced (Supplementary Fig. 35). Furthermore, the fitted energy difference between the S$_1$ and CS states was 12 kJ/mol (this corresponds to a 28 nm shift compared to the S$_1$ energy), which is well in line with the emission data (Fig. 3). It is thus very plausible that the observed long lifetime represents the lifetime of the CS state, illustrating the complex dynamics in these systems.

**Energy of the CS state**

To investigate whether the energy of the CS state is influenced by oligomerization, the redox potentials of the monomer and the dimer were determined in DCM (Supplementary Note 6). Limited solubility in ACN prevented experimental determination of the CS state energy, and we therefore relied on the Rehm-Weller equation for this value. The monomer shows a reversible one-electron oxidation and reduction at +0.71 V and −1.73 V vs. Fc/Fc$^+$, respectively in DCM (Supplementary Fig. 36a, Supplementary Table 4). These values correlate very well with the previously reported redox potentials of the monomer[24], where the reduction was assigned to the BODIPY and oxidation to the anthracene moieties. The dimer in DCM shows two reversible one-electron waves for reduction (Supplementary Fig. 36b, Supplementary Table 4). The first reduction signal at −1.71 V is almost identical to the one observed for the monomer. This agreement indicates that the reduction of the monomeric unit of the dimer is almost identical to that in the monomer. In contrast to the reduction, only one oxidation potential for the dimer was observed close to the end of the solvent window. A small shift of 80 meV for the oxidation potential was seen. It is unlikely that this is caused by anthracene-anthracene interactions, because of their large distance to each other. However, the oxidation peak is not well resolved, and the error margins are therefore large.

Nevertheless, the change in $S_1$ energy is notably larger compared to the change in CS energy (about 160 meV vs 100 meV), and the CS energy change due to oligomerization is small compared to the effect of solvent polarity. It should be noted here that limited solubility of the trimer and tetramer prevented cyclic voltammetry to be conducted.

It is interesting to compare the energy of the CS state measured electrochemically and spectroscopically. The energy of the electrochemically determined CS state is summarized in Supplementary Table 5 and shown schematically in Fig. 3 as a vertical line for the monomer and dimer. It is located at a higher energy as compared to the CS state emission maximum, which is reasonable. Thus, the electrochemically and spectroscopically measured energy of the CS state corroborate each other. In summary, the energy of the CS state is less affected in comparison to the changes seen for the $S_1$ state by oligomerization. Furthermore, it can be tuned separately by the solvent polarity.

## Formation of the triplet state

The ability of the monomer, dimer, trimer, and tetramer to relax from the CS state to the $T_1$ state was assessed by nanosecond transient absorption. The monomer is known to form the triplet state in high yield in both DCM and ACN. Indeed, after excitation to the $S_1$ state, all compounds show both a ground state bleach, as well as an excited state absorption in degassed DCM as well as ACN solutions (Fig. 4a, b). Furthermore, as oxygen is added, all transient absorption features disappear (Supplementary Figs. 37–41). This indicates that all the oligomers populate the triplet state in these solvents. The kinetics of the recovery to the ground state was fitted well assuming first order kinetics, resulting in lifetimes of around 150 µs for the monomer and

generally longer for the oligomers (Table 1, Fig. 4c, d). The triplet lifetime is on the same order of magnitude as iodinated BODIPY[50], and several metal porphyrins. However, what stand out is that the triplet lifetime does not decrease with the number of moieties in the series, despite that the $S_1$ energy decreases. If the reduction of the singlet energy also reduced the triplet energy, a reduction of the triplet lifetime would have been expected from the energy gap law[51–53]. Thus, this feature can be explained by the change in singlet energy being decoupled from the triplet energy.

The yield of intersystem crossing for the monomer is known, and it was therefore used as a reference when determining the yield of intersystem crossing for the oligomers. Figures 4c, d show decays in DCM and ACN, respectively, using absorption and excitation fluence matched conditions (low solubility of the trimer prevented such experiment in ACN). Under such conditions, the yield of intersystem crossing can be determined by comparing the ground state bleach shortly after the excitation pulse. In DCM, the yield of intersystem crossing reduces through the series (Table 1). This is an expected result as the driving force for forming the CS state is continuously reduced through the series. Still, it is rewarding to simulate the yields of emission and intersystem crossing from known measurables in order to gain a thorough understanding of the observed photophysics in the series. A rate equation model was therefore constructed (Supplementary Note 7). It is based on a three-level system, the $S_1$, CS, and $T_1$ states, of which the $S_1$ and CS states are in dynamic equilibrium with each other (Fig. 5a), where the individual rate constants are shown in Supplementary Note 7 and Supplementary Tables 6, 7.

Figure 5b shows the simulated and measured quantum yields of fluorescence and intersystem crossing in DCM. The trends in oligomer

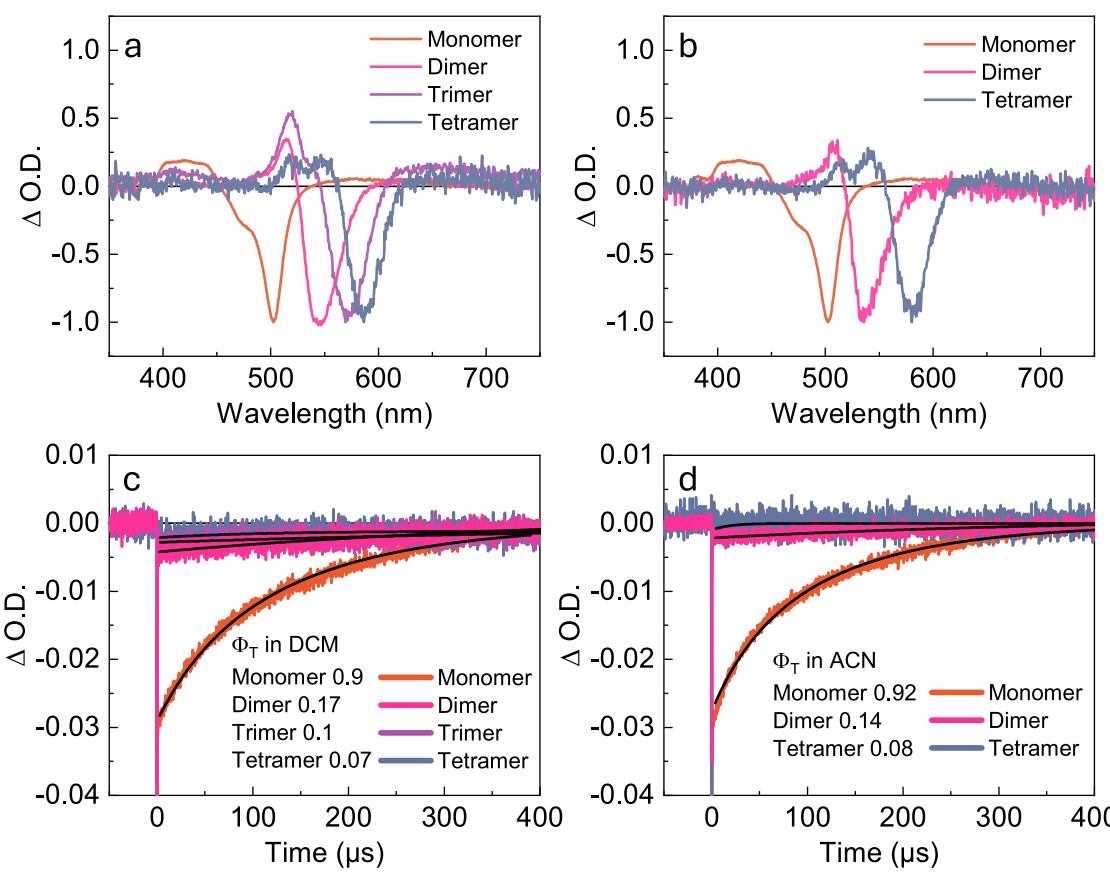

**Fig. 4 | Nanosecond transient absorption.** Normalized transient absorption spectra of the investigated compounds in (**a**) DCM and (**b**) ACN at a time delay of 1 µs obtained with an excitation wavelength corresponding to the absorption maximum of each of the compounds. **c**, **d** Show the ground state bleach kinetics of the monomer, dimer, trimer and tetramer probed at the respective ground state bleach maxima in DCM and ACN, respectively. Limited solubility of the trimer prevented quantification of $\Phi_T$ in ACN. Source data are provided as a Source Data file.

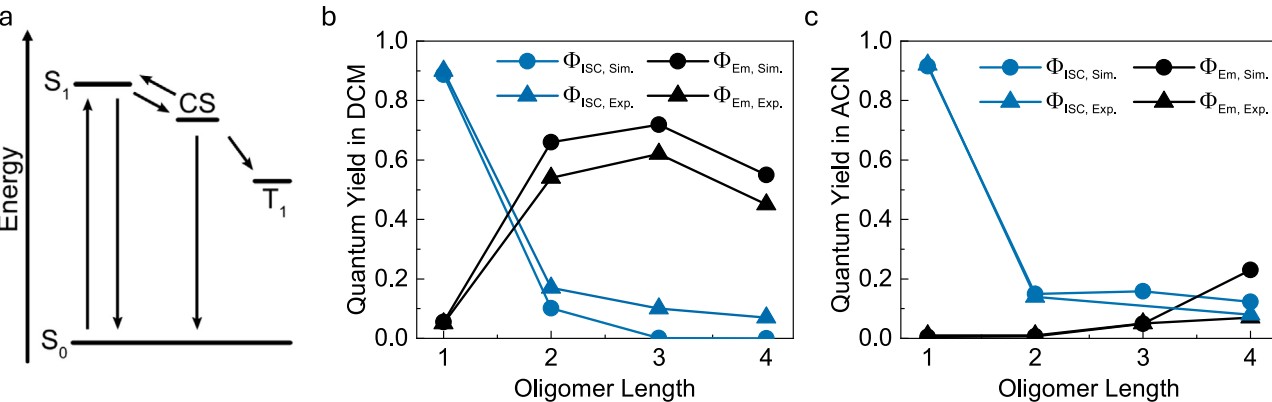

**Fig. 5 | Simulating the photophyiscs with rate equations. a** The kinetic model used to simulate the photophysics of the oligomers. Note that SOCT-ISC entails a direct conversion of the CS state into the $T_1$ state, involving a reverse electron transfer in conjunction with a spin inversion process. **b, c** Simulated (circles) and experimental (triangles) quantum yields of fluorescence (black) and ISC (blue) in DCM (**b**) and ACN (**c**). Source data are provided as a Source Data file.

lengths are captured very well, with a first increasing and then decreasing yield of fluorescence, and a continuously decreasing yield of intersystem crossing. The dependence of the fluorescence yield on oligomer length can be explained by the balance between a decreased driving force for charge separation and an increased non-radiative decay to the ground state. The energetics of the charge separation goes from exothermic for the monomer to endothermic for the oligomers. The result being that the equilibrium concentrations are shifted towards the $S_1$ state with an increased number of moieties in the oligomer. The energetics also explain why a very fast decay component in the TCSPC experiments only could be seen for the monomer (vide supra). Counteracting the shift in the dynamic equilibrium concentration is the non-radiative decay to the ground-state, which increases throughout the series. That internal conversion increases with a reduced excitation energy is commonly observed and is usually explained by the energy gap law[51,52]. We note, however, that for J-aggregates there are examples where a reduced reorganization energy counteracts the effect of the energy gap law[32,54,55]. The simulations also capture the yields of fluorescence and intersystem crossing in ACN solution very well. Here, a high driving force for charge separation results in a low yield of fluorescence throughout the series. A slight increase in emission yield with oligomer length is observed, which can be explained from a reduction in the driving force with oligomer length. Overall, the trends of the fluorescence and intersystem crossing yields with oligomer length is quite straightforwardly explained by the driving force for charge separation. Thus, highlighting the importance of the delicate balance of the energetics between the $S_1$, CS and $T_1$ states for the SOCT-ISC process. Optimization of these energetics could for instance be done by changing the oxidation potential of the anthracene moiety through chemical modifications or by changing the solvent. To demonstrate that solvent polarity can be used as an optimization tool to increase the yield of triplet formation, measurements were conducted on the dimer in benzonitrile, a solvent with a polarity ($\varepsilon = 25.9$) between that of DCM and ACN. As expected, the dimer exhibited an emission quantum yield (29%) that was intermediate between those observed in DCM and ACN. Additionally, the triplet quantum yield (24%; Supplementary Fig. 43), increased by 41% and 71% compared to DCM and ACN, respectively. Also, although our simulations capture the overall trend well, the absolute values are less well explained. We here note that it is possible to perform excellent fits with our model, by for instance using the energy of the CS state, the rate of charge recombination to the $T_1$ state, or the rate of charge separation as fitting parameters. However, as we do not know which of these parameters changes most, we stop at the point where approximated rates fit the overall trends in yields well.

The yields of intersystem crossing and fluorescence are relatively low for the oligomers in ACN. This is attributed to an increase in the non-radiative rate constant in ACN for the CS state, which is well explained by Marcus theory for electron transfer. To investigate the rate changes throughout the series, we conducted femtosecond transient absorption experiments (which corroborated the time-resolved emission experiments) and analyzed the photophysics using Marcus theory (Supplementary Note 8, Supplementary Figs. 44–46). The initial formation of the CS state as well as its relaxation to the triplet and ground states are electron transfer processes. The rate of electron transfer reactions normally increase steeply with the driving force until the driving force equals the reorganization energy (so-called Marcus normal region). With further increase in the driving force, the rate decreases sharply (so-called Marcus inverted region). Redox potentials for the monomer and dimer could be obtained (vide infra), and these were used to analyse the rate for charge recombination to the triplet and ground states. The obtained rates align with the predictions from Marcus theory (Supplementary Fig. 46). The rate of recombination to the triplet state increases with increasing driving force, with the monomer exhibiting a larger driving force and consequently a higher rate. Conversely, the rate of recombination to the ground state decreases with increasing driving force, meaning the monomer, having a larger driving force, shows a slower recombination rate to the ground state. This combination gives the monomer the prerequisites for forming triplets more efficiently than the oligomers, explaining the short lifetime observed for the CS state of the oligomers. Thus, the increase in the non-radiative rate with oligomer length is not a consequence of oligomerization as such, but rather of the mechanism through which the triplet state is reached. Furthermore, the recombination from the CS to the ground state is likely to be in the inverted Marcus regime for any molecular system. High efficiencies for the SOCT-ISC mechanism are therefore only feasible at considerable CS state energies because the unwanted recombination to the ground state becomes slower the higher the CS state energy. Thus, limiting this mechanism to energies roughly equaling the visible part of the electromagnetic spectrum. It is important to note that the low yields, attributed to fast ground state recombination in the oligomers, could have other explanations beyond the one suggested by the analysis based on Marcus theory. However, investigating these alternative explanations is beyond the scope of this work.

## Discussions

Exciton coupling is shown to selectively stabilize the first excited singlet state energy, without significantly perturbing the energy of the

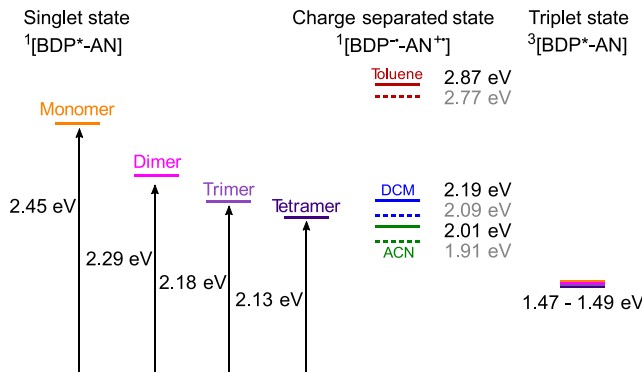

**Fig. 6 | A summary of the effect of oligomerization.** Schematic energy diagram, illustrating the lowered $S_1$ energy levels of the dimer, trimer and tetramer. The energies of the singlet states were taken as the peak maximum of the absorption spectra (Fig. 2b). The energies of the charge separated states were estimated from the redox potentials and are shown for the monomer in solid lines and dimer in dotted lines (Supplementary Table 5). The energies of the triplet states were taken from quantum mechanical calculations (Fig. 2b).

first excited triplet state. This approach thus enables a route to significantly reduce energy losses for triplet sensitizers. We show this in a so-called SOCT-ISC sensitizer by synthesizing a set of new oligomeric BODIPY-anthracene dyads. The design principles in the coupling of dyad moieties focused on the transition dipole moments being aligned in a head-to-tail fashion, but steric hindrance breaking the conjugated system. The oligomers show a gradual lowering of the $S_1$ energy with length (Fig. 6). The dimer shows a shift of 36 nm, the trimer 63 nm and the tetramer 77 nm compared to the monomer. This, as well as the increasing molar absorptivity with oligomer length indicates exciton coupling in the system. Thus, the shift in absorption confirms a successful shift of the $S_1$ state energy to longer wavelengths.

The energy of the charge separated state, where the BODIPY and anthracene units are reduced and oxidized, respectively, is highly dependent on solvent polarity. The choice of solvent therefore modulates the driving force for charge separation from the $S_1$ state. Because of the lower energy of the $S_1$ state in the oligomers compared to the monomer, a solvent with higher polarity was needed to quench the fluorescence for the oligomers. The dynamics of the $S_1$-CS state interaction was examined in detail for the monomer. It was found that a model that included a dynamic equilibrium between these two states were needed to explain the photophysical properties. Recombination from the CS to the ground state was shown to be in the inverted Marcus regime, practically limiting the SOCT-ISC mechanism to relatively high energies. Electrochemical analysis was used to complement spectroscopical investigation and showed that the reduction and oxidation potentials of the dimer do not change to a great extent as compared to the monomer.

The ability of all oligomers to populate the triplet state was shown by nanosecond transient absorption spectroscopy, and the experimentally obtained yields could successfully be simulated by rate equations. Importantly, the corresponding kinetic traces showed no decrease in the triplet lifetime when increasing the oligomer length. That the lifetime of a molecular series does not reduce with decreasing excitation energy is unusual but can be explained by the triplet energy being constant in the series, which was confirmed by quantum mechanical calculations. The conserved triplet lifetime highlights a secondary beneficial feature of using exciton coupling for decreasing the excitation energy of a sensitizer. This is important since long triplet lifetimes are an essential property of photo sensitizers.

We suggest the following design concepts to optimize the photophysical performances beyond those presented herein. Electron delocalization is here restricted through steric bulk. This method is easily synthetically implemented but allows for a distribution of dihedral angles between monomers (thus a small but variable amount of electron delocalization), resulting in broader than expected absorption and emission spectral envelopes. By locking the conformation between monomers in an orthogonal orientation, for instance through the introduction of spiro-carbons, electron delocalization is minimized, and a significant narrowing of the absorbance and emission spectra is expected. Spiro-carbons have previously been introduced in for instance fluorene to form spirobifluorene, which crystal structure shows perpendicular directions of the two fluorene moieties in the dimer[56]. The benefits of the SOCT-ISC mechanism are that the lifetime of the acquired triplet state is long, and that the method works in absence of heavy atoms. The drawback is that it depends on an intermediate CT state, which energy is highly solvent polarity dependent[18]. Furthermore, in exciton coupled systems, the driving force for reaching the triplet state is reduced at the same time as the rate of charge recombination to the ground state is increased, lowering the yield of triplet formation. However, the yields of emission are high and lifetimes relatively long for all oligomers in toluene solution, indicating that exciton coupling itself does not significantly deactivate the excited state. Thus, limitations for the SOCT-ISC process can most likely be overcome by the utilization of other methods of increasing ISC, for instance through the incorporation of heavy atoms.

In summary, exciton coupling was used to selectively lower the energy of the $S_1$ state without affecting the triplet state energy to a great extent. This intramolecular exciton-exciton coupling strategy can be a useful addition in the design of heavy atom free photosensitizers and dyes for light emitting diodes with a small singlet-triplet energy gap.

## Methods

**Synthesis:** All reactions were carried out under ambient conditions unless stated differently, for example performed under $N_2$ atmosphere. Glassware were oven dried prior to use, unless indicated otherwise. Common reagents, solvents, or materials were obtained from Sigma-Aldrich Chemical Co. and used without further purification. Dry solvents for reactions sensitive to moisture and/or oxygen were obtained through a solvent purifying system (inert PureSolv-MD-5). Column chromatography was performed using silica gel (VWR 40 to 63 μm) unless stated otherwise. Preparative thin layer chromatography was performed on silica gel plates (Uniplate_TM, UV254 indicator, 20×20 cm, 2000 micron, Miles Scientific, purchased from Sigma-Aldrich Chemical Co). Size exclusion column chromatography (SEC) was performed using Bio-Beads S-X3 Support as a stationary phase and DCM as eluent. NMR spectra ($^1$H, $^{13}$C, $^{19}$F) were recorded on a BRUKER Avance III spectrometers (600/700/800 MHz $^1$H; 151/176/201 MHz $^{13}$C; 564/659/753 MHz $^{19}$F). The 700 MHz spectrometer is equipped with an AVIII console and a 5 mm QCI H/F–C/N–D–05 Z cryo–probe. The 600 MHz spectrometer is equipped with a NEO console and a QCI H&F/P/C–N–D–05 Z XT cryo–probe. Spectra were taken in CDCl$_3$ (containing tetramethylsilane with 0.00 ppm as an internal reference) or CD$_2$Cl$_2$ as solvent. Coupling constants ($J$ values) are given in Hertz (Hz) and chemical shifts are reported in parts per million (ppm). $^{13}$C spectra are decoupled from $^1$H. High-resolution MS was obtained from an Agilent 1290 infinity LC system equipped with an auto sampler in tandem with an Agilent 6520 Accurate Mass Q-TOF LC/MS. LDI MS data were collected with a Rapiflex MALDI ToF/ToF mass spectrometer (Bruker Daltonics, Bremen, Germany). The instrument is equipped with a Smartbeam 3D UV laser (355 nm) and operated in reflector-positive ionization mode. A number of 200 shots were acquired at 1 kHz with a mass range set to m/z 400–3200. Melting points were measured using a BÜCHI Melting Point B-545 instrument. IR spectra were recorded using an INVENIO R FT-IR from BRUKER. Analytical HPLC was performed on a Thermo scientific, Dionex Ultimate 3000 system equipped with a Dionex Ultimate 3000 Variable

Wavelength Detector and a Reverse Phase Dionex Acclaim Polar-Advantage II column. The solvent used was a mixture of MilliQwater and acetonitrile, both containing 0.1% formic acid. The oligomers (Monomer, Dimer, Trimer1, Trimer2 and Tetramer) were run using the $\lambda_{max}$ of the regarding compound as the detection wavelength. The compounds were run using a gradient ramping from 5% to 95% MeCN over a time period of 15 min with afterward holding of the final ratio for 10 min in the case of Monomer, dimer, trimer1 and trimer2. The tetramer was run using a gradient ramping from 5% to 95% MeCN over a time period of 10 min with afterward holding of the final ratio for 20 min. Purities are given in the caption of Supplementary Figs. 2, 6, 11, 14, and 17.

Optical spectroscopy: Absorption spectra were measured using a spectrophotometer (LAMBDA 950, PerkinElmer). Steady-state emission spectra, excitation spectra and emission lifetimes were measured with a spectrofluorometer (FLS1000, Edinburgh Instrument) and are corrected using the emission correction files provided by the manufacturer. The emission quantum yields were calculated using the relative method, using a standard (Fluorescein) with known emission quantum yield[57]. Fluorescein in 0.1 M NaOH ($\Phi_f = 0.91$) was used as a reference compound for $\Phi_{Em}$ determination[57]. (Excitation at 480 nm, Refractive index: 1.33). The samples were measured in solution at 22 °C, refractive indices: toluene = 1.497, DCM = 1.4125, ACN = 1.3405. For the emission lifetime measurements, the samples were excited by a 375 nm or 510 nm picosecond pulsed diode laser (Edinburgh Instruments).

Nanosecond transient absorption measurements were performed on an Edinburgh Instrument LP 980 spectrometer equipped with an ICCD (Andor). A Spectra-Physics Nd:YAG 532 nm laser (pulse width ~7 ns) coupled to a Spectra-Physics primoscan optical parametric oscillator (OPO) was used as pump source. The samples used for transient absorption measurements were prepared in a glovebox using anhydrous and degassed solvents. The samples used for steady-state absorption and emission spectra were prepared using anhydrous solvents (including molar absorptivities, fluorescence decays and fluorescence quantum yield). Anhydrous solvents were necessary since the CS state was quenched by the presence of even small amounts of water. There are literature reports showing that the charge separated state lifetime can be heavily influenced by hydrogen bonding[58]. However, such interactions are not possible in our system. The quenching mechanism involving water was not investigated further since it is beyond the scope of this report. In intense laser light the concentration of excited state species might become high enough to allow bimolecular reactions. For triplet states, triplet-triplet annihilation might occur if two excited molecules collide leading to increased deactivation rate of the excited state population[59]. The time dependence of the triplet excited state concentration (if the triplet formation occurs within the instrument response function of the instrument) is given by:

$$\frac{d[^3A^*]}{dt} = -2k_{TTA}\left[^3A^*\right]^2 - k_T\left[^3A^*\right] \tag{1}$$

where $k_T$ and $k_{TTA}$ are the first order intrinsic and second order triplet-triplet annihilation rate constants, respectively. Equation 1 has an analytical solution giving the time dependent triplet excited state concentration.

$$\left[^3A^*\right] = \left[^3A^*\right]_0 \frac{1-\beta}{e^{k_T t} - \beta}, \text{ where } \beta = \frac{2k_{TTA}\left[^3A^*\right]_0}{k_T + 2k_{TTA}\left[^3A^*\right]_0} \tag{2}$$

The parameter $\beta$ describes the ratio of second order decay over total decay rate of the triplet state. Equation 2 was used to fit the nsTA data of the monomer measured at a series of different intensities in a global fit to obtain the triplet lifetime.

Femtosecond transient absorption (fsTA) measurements were performed with a Ti:sapphire oscillator (Mai-Tai, Spectra Physics) which was used as seed to a regenerative amplifier (Solstice Ace, Spectra Physics) pumped by a frequency-doubled diode-pumped Nd:YLF laser (Ascend, Spectra Physics). This produced pulses of around 60 fs duration (FWHM) at a 1 kHz repetition rate. The 800 nm output from the amplifier was split, and the two beams were used as pump and probe light. To achieve the desired excitation wavelength of 390 nm (with an energy of 1 µJ per pulse at the sample), we employed an optical parametric amplifier (TOPAS PRIME, Light Conversion Ltd.) The probe light was directed onto a translating CaF2 plate to generate a supercontinuum, while the pump beam's timing was adjusted relative to the probe beam using an optical delay stage (with a range from 0 to 10 ns). The supercontinuum was split into a probe and reference beam, and the probe beam overlapped with the pump at the sample. The transmitted probe and reference beam were directed to optical fibers and detected by a CCD camera (iXon-Andor) operating synchronously with the 1 kHz laser. The transient spectra were obtained from the difference of the probe light divided by the reference with and without excitation of the sample by the pump beam; 2000 spectra were averaged per delay time using a custom LabVIEW program controlling the setup.

The cyclic voltammetry measurements were performed with a CHI-potentiostat controlled using CHI650A software (version 11.15). Platinum electrodes were used as the working and counter electrodes and Ag/AgCl in saturated KCl was used as the reference electrode. The measurements were performed in thoroughly degassed (using Argon) dichloromethane (DCM) with 0.1 M tetra-*n*-butylammonium perchlorate TBAPF$_6$ (Sigma–Aldrich) at a scan rate of 0.1 V/s. Ferrocene/Ferrocenium (Fc/Fc$^+$) was used as an external standard with E$_{1/2}$ at approximately 0.61 V vs. Ag/AgCl in DCM.

Quantum mechanical calculations were performed using Gaussian 16, revision B.01[47,48]. The geometry of singlet and triplet ground states were optimized in vacuum at the ωB97XD/6-31 g(d) level of theory and singlet excitation energies were obtained through TD-DFT using the singlet ground state geometry. The energy difference between the ground singlet and triplet states therefore represents adiabatic transitions, whereas the energy difference between the ground and excited singlet states represents vertical transitions.

## Data availability
The data that support the findings of this work are available within the Article and its Supplementary Information files. Source data are provided with this paper.

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

## Acknowledgements

The Swedish NMR center is acknowledged for access to high-field NMRs. The authors especially acknowledge Zoltan Takacs for performing diffusion NMR spectroscopy and analysis thereof. K.B. gratefully acknowledge financial support from the Knut and Alice Wallenberg Foundation (KAW 2017,0192). B.A. acknowledges the Swedish research council (VR) for support. Our research relied on computational resources provided by the National Academic Infrastructure for Supercomputing in Sweden (NAISS) at C3SE and NSC partially funded by the Swedish research council through grant agreement no. 2022-06725.

## Author contributions

K.B. together with C.S. designed the project idea and the molecules. C.S. synthesized and analyzed the molecules, and performed absorption, emission and initial transient absorption experiments and analysis thereof. R.R. performed electrochemical analysis of the compounds as well as transient absorption measurements. M.R. performed DFT calculations. B.A., K.B., R.R., and C.S. analyzed all collected results. J.H. performed MALDI analysis of the synthesized BODIPY oligomers. All authors contributed to writing the manuscript. All authors have given approval to the final version of the manuscript.

## Funding

## Competing interests

The authors declare no competing interest.
