## [Peer Review File · Nature Communications]

Lowering of the singlet-triplet energy gap via intramolecular exciton-exciton couplingReviewer #1 (Remarks to the Author):

In this study, Borjesson and colleagues describe the properties of several donor-acceptor BODIPY oligomers and propose a method for tuning the singlet-triplet (S-T) energy gap via exciton coupling interaction. The authors chemically linked multiple BODIPY fragments and observed that exciton coupling between the chromophores significantly influences the first excited singlet state energy (lowering it), while the energy of the lowest triplet excited state remains largely unaffected. This observation is attributed to the dependence of exciton coupling strength on the magnitude of the transition dipole moments. These findings are pertinent to the photochemistry community, and the approach holds potential for devising innovative photosensitizers.

However, it is noteworthy that the novelty of the proposed approach is constrained. The decrease in the S-T gap and enhanced intersystem crossing via exciton coupling were previously suggested for a series of BODIPY dimers by Flamigni and colleagues in 2009. Regrettably, this prior work is not referenced in the current manuscript, and it is possible that the authors are unaware of it (see doi:10.1039/B813638F for details).

Another concern pertains to the effectiveness of this approach in promoting ISC. The presented data indicates that, despite lowering the S-T gap, ISC is inefficient for trimer and tetramer molecules due to an increase in ground state recombination. Consequently, this strategy may only be effective for dimers. Moreover, the achieved values of ISC quantum yields for the dimer molecules are moderate (14-17%).

While the conceptual foundation of this work is intriguing, it is my opinion that the authors have not fully exploited its potential. In terms of novelty and significance, I believe the manuscript falls short of the journal's expectations.

Additionally, I must highlight some issues with the presented data:

The NMR data for the trimer and tetramer clearly indicates the presence of impurities. I recommend utilizing size-exclusion chromatography to purify these compounds.

The absorption spectra depicted in Figures S22-S26 exhibit a baseline problem, appearing noisy in the 600-700 nm region.

The size of figures presented in the Supplementary Information is too small.

Reviewer #2 (Remarks to the Author):

This manuscript reported a method to decrease the S1 state energy, while keep the T1 state energy intact. The energy loss can be minimized with this method. And it impose significant effect in a few aspects, such as to maximize the anti-Stokes shift in triplet-triplet annihilation upconversion.

I suggest acceptance of the manuscript after revision. The T1 state energy is based on theoretical computation. Since this is one of the critical photophysical parameters discussed in this manuscript, I suggest the T1 state energy level to be confirmed with experimental methods, such as the intermolecular triplet energy transfer (by nanosecond transient absorption).

Reviewer #3 (Remarks to the Author):

The paper presents an interesting idea on how to decrease the energy gap between the lowest singlet (S1) and triplet (T1) states of an organic compound. The aim is to bring S1 lower without affecting T1, so that less excitation energy is required to populate the T1 state.

When S1 and T1 involve the same singly two occupied orbitals (in a zero order approximation), the energy gap is given by the exchange integral of these two orbitals. This integral is small for $\pi\pi^*$ transitions or charge transfer (CT) states. However, in both cases spin-orbit coupling (SOC) between S1 and T1 is small, making both forward and reverse ISC slow. ISC can be enhanced by bringing the triplet of a $\pi\pi^*$ transition just below the CT triplet. The corresponding $\pi\pi^*$ singlet will have large oscillator strength but will not be S1. Hence this configuration (here called SOCT-ISC) is useful for large triplet yields.

The authors propose a strategy for lowering the optically allowed singlet state: Excitonic coupling between the singlet states of monomer units in a polymer will lead to splitting of these states that increases with the oscillator strength of the monomer transitions. When the monomer units are arranged such that the transition dipoles of the monomers are in head-to-tail fashion, the complete oscillator strength of the excitonic states goes to the lowest state. Since the corresponding triplet states have zero oscillator strength, their excitonic states will mostly remain degenerate. I.e., the S1-T1 gap will be reduced.

In order to demonstrate their new concept, the authors synthesize oligomers of a BODIPY-anthracene dyad which has been studied before for the enhancement of ISC by the CT-mechanism sketched above. The BODIPY units are connected in head-to-tail fashion with respect to the S0-S1 transition dipole. Bulky substituents lead to orthogonal orientation of the sequence of π -electron systems, i.e. the π -system is not enlarged.

These systems show the expected red shift of the strong absorption band with increasing oligomer size, which is also reflected by a red-shift of the fluorescence emission. The ISC remains small unless the CT state of the BODIPY-anthracene dyad is shifted below the excitonic state by solvent effect. This becomes increasingly difficult with increasing the oligomer size, since the energy of the CT state is not affected by excitonic interaction. I conclude that these oligomer systems are actually less useful for efficient ISC.

The study appears to me as an exercise in the manipulation of strong absorption bands with an interesting design idea. However, does this concept really compete well with the alternative strategy of increasing the π -electron delocalization? Here the open question in my opinion, and an interesting subject for further study, is the effect of the increased delocalization on the exchange integral. I.e., will the S-T gap become smaller or larger? Such a comparison could reveal a possible advantage of the new idea.

In summary, I think this is an interesting study that brings more insight into possible tools for designing organic molecules with desired singlet/triplet spacings. On the other side, the molecules that were actually studied do not improve the desired properties (triplet yield) compared to the monomer. The article should be publishable (after considering the comments given below). However, whether it is suitable for this journal or better for a more specific journal should be decided by the journal policy.

Specific comments:

According to table 1, the emission yield of the monomer decreases from 80% to 5% when toluene is replaced by DCM as solvent. The CT emission was subtracted, i.e., the yield should refer to the LE state. Since the corresponding oscillator strength should not be affected by the solvent, one should expect that the fluorescence yield is lowered by quenching, and hence the lifetime should also decrease by a factor of 20. However, the lifetime is almost unaffected. I conclude that the measured lifetime must refer to different electronic states in toluene and

DCM, presumably the locally excited state (LE) in toluene and the CT state in DCM. I should then conclude that the oscillator strength of the CT is about 5% that of the LE. The authors indicate the wavelength of excitation, but not that of the detection.

One can make a similar observation for the dimer, trimer, and tetramer: In going from toluene to DCM as solvent the emission quantum yields drops, but the lifetime even increases! In these cases, no correction for a CT emission is mentioned in the table caption, and only a single emission maximum is given. Fig. 3 shows CT emission only for the monomer (with a second maximum) and dimer (as a shoulder). Excluding the red tail of the emission from lifetime measurements should give a clearer picture.

Page 2, line 10: The sentence breaks off after “regardless”.

Page 7, near bottom: The energy gap law says that the Franck-Condon weighted density of states (FCWD) decays nearly exponential with the energy gap for a given system. I.e., the FCWD for two molecules that have a different number of normal vibrations are by no means equal at the same value of the energy gap. Hence, the energy gap law can not “explain” a decrease of a transition rate by an increased energy gap in comparing two different molecules.

Reviewer #4 (Remarks to the Author):

In the present manuscript the authors propose a strategy to shift the singlet excited states of photosensitizers by forming an excitonic J-aggregates while keeping the energy of a triplet state constant. The strategy is demonstrated experimentally on a series of covalently linked bodipy-anthracene dyes.

The work is in my opinion suitable for publication in Nature Communications. However, prior to the publication the authors should address the following issues:

1. J-aggregation as a strategy for shifting the singlet state energy without affecting the triplet state is not new and has been reported in the literature as early as in 1939. In the paper by Kautsky and Merkel published in *Naturwissenschaften*, 27, 195 (1939) the authors report that the aggregation of dyes facilitates their action as photophysical sensitizers in photochemical reactions. In fact, this work has motivated later Kasha's work on exciton theory whose initial motivation was to explain enhanced triplet generation in J-aggregates. This old seminal work should be clearly acknowledged.
2. The spectra in Fig 1. clearly show the shift of the excitation energy which is typical for J-aggregates. However, I do not see clearly the exchange narrowing effect that should be also present. The authors might try to quantify the widths of the bright exciton state and compare the results with the prediction of the exciton model in order to see if the trend is correctly reproduced.
2. I am puzzled by the fact that TDDFT calculations based on the long-range corrected functional so strongly overestimate the energy shift in Fig. 1b. This might be due to a rather small basis set used (6-31g(d)). I think that calculations using a triple-zeta basis set should be feasible for this systems and the authors might try to perform them in order to improve the agreement.
3. It would interesting to provide pictures of the electronic transition densities for the bright

states.

4. The authors mention the finding that for J-aggregates reduced reorganization energy counteracts the effect of the energy gap law citing their own work from 2021. This fact has been first theoretically predicted by Scharf and Dinur in 1984 (Chem. Phys. Lett., 105, 78 (1984)). Moreover, the modified energy gap law for J-aggregates has been developed and the size dependence of the radiative and non-radiative rates in J-aggregates has been predicted in J. Phys. Chem. A, 124, 10143 (2020). These works should be properly acknowledged.

Altogether I find the finding in the present paper sufficiently interesting and novel to warrant the publication in Nat. Comm.

Reviewer #1:

In this study, Borjesson and colleagues describe the properties of several donor-acceptor BODIPY oligomers and propose a method for tuning the singlet-triplet (S-T) energy gap via exciton coupling interaction. The authors chemically linked multiple BODIPY fragments and observed that exciton coupling between the chromophores significantly influences the first excited singlet state energy (lowering it), while the energy of the lowest triplet excited state remains largely unaffected. This observation is attributed to the dependence of exciton coupling strength on the magnitude of the transition dipole moments. These findings are pertinent to the photochemistry community, and the approach holds potential for devising innovative photosensitizers.

However, it is noteworthy that the novelty of the proposed approach is constrained. The decrease in the S-T gap and enhanced intersystem crossing via exciton coupling were previously suggested for a series of BODIPY dimers by Flamigni and colleagues in 2009. Regrettably, this prior work is not referenced in the current manuscript, and it is possible that the authors are unaware of it (see doi:10.1039/B813638F for details).

Thank you. There is indeed several BODIPY dimers in the literature. In our original manuscript we only cited BODIPY dimers that were beta-tethered, in the updated manuscript we also include citations to work connecting BODIPYs in other geometries (including the work by Flamigni). The updated text reads:

“Dimerization of BODIPY dyes has previously been achieved in the meso, α , and β positions.^{39, 40, 41} Of these attachment positions, β -tethering also leads to oligomerization of BODIPY dyes.^{42, 43, 44}”

The work by Flamigni et al mentioned above describes BODIPY dimers attached in the beta position. The transition dipole moment of the two BODIPYs are close to be in an orthogonal orientation, leading to a medium sized exciton coupling and a split of the lowest energy transition into two new bands (one lower and one higher in energy of about equal magnitude). The prime experimental observation of this work is that the yield of intersystem crossing is increased in the dimers, and the authors hypothesize that this is due to a reduced triplet singlet gap. In later works it has been shown that the increased yield of intersystem crossing is rather due to the orthogonality of the transition dipole moments of the BODIPY moieties (Angew. Chem. Int. Ed. 2011, 50, 11937-11941).

Another concern pertains to the effectiveness of this approach in promoting ISC. The presented data indicates that, despite lowering the S-T gap, ISC is inefficient for trimer and tetramer molecules due to an increase in ground state recombination. Consequently, this strategy may only be effective for dimers. Moreover, the achieved values of ISC quantum yields for the dimer molecules are moderate (14-17%).

While the conceptual foundation of this work is intriguing, it is my opinion that the authors have not fully exploited its potential. In terms of novelty and significance, I believe the manuscript falls short of the journal's expectations.

We agree with the reviewer that the yield of triplet formation is of importance. We have therefore examined the yield of triplet formation in a solvent having a polarity in between that of DCM and acetonitrile. This to show that it is possible to optimize the yield of triplet formation by the choice of solvent. Indeed, our new results show a significant increase in the yield of triplet formation for the dimer (an increase by 41% and 71% compared to DCM and ACN, respectively), demonstrating the potential of using solvent polarity as a means to optimize the yield of triplet formation. A new section describing this new result has been added in the result section, it reads:

“Overall, the trends of the fluorescence and intersystem crossing yields with oligomer length is quite straightforwardly explained by the driving force for charge separation. Thus, highlighting the importance of the delicate balance of the energetics between the S_1 , CS and T_1 states for the SOCT-ISC process. Optimization of these energetics could for instance be done by changing the oxidation potential of the anthracene moiety through chemical modifications or by changing the solvent. To demonstrate that solvent polarity can be used as an optimization tool to increase the yield of triplet formation, measurements were conducted on the dimer in benzonitrile, a solvent with a polarity ($\epsilon = 25.9$) between that of DCM and ACN. As expected, the dimer exhibited an emission quantum yield (29%) that was intermediate between those observed in DCM and ACN. Additionally, the triplet quantum yield (24%; Supplementary Figure 43), increased by 41% and 71% compared to DCM and ACN, respectively.”

Furthermore, we have added a discussion on design strategies that could be used to enhance the photophysical properties of exciton coupled oligomers. This new paragraph is located in the conclusion section, and it reads:

“Based on this work, the following conclusions on further design concepts to optimize the photophysical performances can be drawn. Electron delocalization is here restricted through steric bulk. This method is easily synthetically implemented but allows for a distribution of dihedral angles between monomers (thus a small but variable amount of electron delocalization), resulting in broader than expected absorption and emission spectral envelopes. By locking the conformation between monomers in an orthogonal orientation, for instance through the introduction of spiro-carbons, electron delocalization is minimized, and a significant narrowing of the absorbance and emission spectra is expected. The benefits of the SOCT-ISC mechanism are that the lifetime of the acquired triplet state is long, and that the method works in absence of heavy atoms. The drawback is that it depends on an intermediate CT state, which energy is highly solvent polarity dependent. Furthermore, in exciton coupled systems, the driving forces for reaching the triplet state is reduced at the same time as the rate of charge recombination to the ground state is increased, lowering the yield of triplet formation. However, the yields of emission are high and lifetimes relatively long for all oligomers in toluene solution, indicating that exciton coupling itself does not significantly deactivate the excited state. Thus, limitations for the SOCT-ISC process can be overcome by the utilization of other methods of increasing ISC, for instance through the incorporation of heavy atoms.”

Additionally, I must highlight some issues with the presented data:

The NMR data for the trimer and tetramer clearly indicates the presence of impurities. I recommend utilizing size-exclusion chromatography to purify these compounds.

The use of SEC as a purification method is a good idea, and it was already done twice for these compounds. However, SEC was not mentioned in the methods section of the main part of the manuscript. A note has been added in the method section, it reads:

“Size exclusion column chromatography (SEC) was performed using Bio-Beads S-X3 Support as a stationary phase and DCM as eluent.”

The absorption spectra depicted in Figures S22-S26 exhibit a baseline problem, appearing noisy in the 600-700 nm region. The size of figures presented in the Supplementary Information is too small.

We would like to thank the reviewer for highlighting the baseline problem and the small size of figures. The figures in the Supplementary Information have been enlarged, and spectra with baseline artefacts have been replaced with better quality ones.

Reviewer #2:

This manuscript reported a method to decrease the S1 state energy, while keep the T1 state energy intact. The energy loss can be minimized with this method. And it impose significant effect in a few aspects, such as to maximize the anti-Stokes shift in triplet-triplet annihilation upconversion. I suggest acceptance of the manuscript after revision. The T1 state energy is based on theoretical computation. Since this is one of the critical photophysical parameters discussed in this manuscript, I suggest the T1 state energy level to be confirmed with experimental methods, such as the intermolecular triplet energy transfer (by nanosecond transient absorption).

We agree with the reviewer that the energy of the triplet states is very important for the conclusions in this manuscript. During the revision process we have tried two new methods to validate the triplet energies. The first one was experimental and is the most direct method that we could think of. The monomer was dissolved in m-THF, an iodine-based salt was added to enhance ISC through the heavy atom effect, and the solution was frozen to 100 K. However, no phosphorescence could be detected. This result might not be very surprising as organic dyes generally have a very small transition dipole moment between the S0 and T1 states, but it was worth to try.

Furthermore, we have validated the triplet energies by repeating our calculations in a solvent environment (DCM; the calculations in our manuscript were conducted in vacuum). No differences in triplet energies could be seen throughout the series (1.49 eV for all). See Reviewer 4, question 2 for an elaborate discussion on calculations.

Reviewer #3:

The paper presents an interesting idea on how to decrease the energy gap between the lowest singlet (S1) and triplet (T1) states of an organic compound. The aim is to bring S1 lower without affecting T1, so that less excitation energy is required to populate the T1 state.

When S1 and T1 involve the same singly two occupied orbitals (in a zero order approximation), the energy gap is given by the exchange integral of these two orbitals. This integral is small for npi^* transitions or charge transfer (CT) states. However, in both cases spin-orbit coupling (SOC) between S1 and T1 is small, making both forward and reverse ISC slow. ISC can be enhanced by bringing the triplet of a pi^*pi^* transition just below the CT triplet. The corresponding pi^*pi^* singlet will have large oscillator strength but will not be S1. Hence this configuration (here called SOCT-ISC) is useful for large triplet yields.

The authors propose a strategy for lowering the optically allowed singlet state: Excitonic coupling between the singlet states of monomer units in a polymer will lead to splitting of these states that increases with the oscillator strength of the monomer transitions. When the monomer units are arranged such that the transition dipoles of the monomers are in head-to-tail fashion, the complete oscillator strength of the excitonic states goes to the lowest state. Since the corresponding triplet states have zero oscillator strength, their excitonic states will mostly remain degenerate. I.e., the S1-T1 gap will be reduced.

In order to demonstrate their new concept, the authors synthesize oligomers of a BODIPY-anthracene dyad which has been studied before for the enhancement of ISC by the CT-mechanism sketched above. The BODIPY units are connected in head-to-tail fashion with respect to the S0-S1 transition dipole. Bulky substituents lead to orthogonal orientation of the sequence of pi-electron systems, i.e. the pi-system is not enlarged.

These systems show the expected red shift of the strong absorption band with increasing oligomer size, which is also reflected by a red-shift of the fluorescence emission. The ISC remains small unless the CT state of the BODIPY-anthracene dyad is shifted below the excitonic state by solvent effect. This becomes increasingly difficult with increasing the oligomer size, since the energy of the CT state is not affected by excitonic interaction. I conclude that these oligomer systems are actually less useful for efficient ISC.

The study appears to me as an exercise in the manipulation of strong absorption bands with an interesting design idea. However, does this concept really compete well with the alternative strategy of increasing the pi-electron delocalization? Here the open question in my opinion, and an interesting subject for further study, is the effect of the increased delocalization on the exchange integral. I.e., will the S-T gap become smaller or larger? Such a comparison could reveal a possible advantage of the new idea.

In summary, I think this is an interesting study that brings more insight into possible tools for designing organic molecules with desired singlet/triplet spacings. On the other side, the molecules that were actually studied do not improve the desired properties (triplet yield) compared to the monomer. The article should be publishable (after considering the comments given below). However, whether it is suitable for this journal or better for a more specific journal should be decided by the journal policy.

We are happy to read that the reviewer thinks our concept is interesting.

Regarding how well the concept competes with other strategies (such as HOMO-LUMO spatial separation), we are currently implementing the strategy described in this work on so called TADF dyes. This with the hypothesis that these two strategies of lowering the S1-T1 energy can be implemented into the same molecular framework.

Regarding that we do not improve the desired property (triplet yield). We do not agree here with the reviewer that the triplet yield is the most important property in this manuscript. Our main claim is that we are modifying the singlet – triplet energy difference. As the monomer was chosen based on

its very high triplet yield (close to unity), it would be very hard to improve. However, in the updated manuscript we have examined if it is possible to improve the triplet yield of the dimer by changing solvent polarity, and it is possible (see answer to comment by reviewer 1). Thus, demonstrating how to optimize the yield in this kind of systems.

Specific comments:

According to table 1, the emission yield of the monomer decreases from 80% to 5% when toluene is replaced by DCM as solvent. The CT emission was subtracted, i.e., the yield should refer to the LE state. Since the corresponding oscillator strength should not be affected by the solvent, one should expect that the fluorescence yield is lowered by quenching, and hence the lifetime should also decrease by a factor of 20. However, the lifetime is almost unaffected. I conclude that the measured lifetime must refer to different electronic states in toluene and DCM, presumably the locally excited state (LE) in toluene and the CT state in DCM. I should then conclude that the oscillator strength of the CT is about 5% that of the LE. The authors indicate the wavelength of excitation, but not that of the detection.

We agree that this is a surprising observation (and it should be noted that two previous publications working with the monomer has observed the same phenomena but not tried to explain it). Our explanation is based on a reversibility for the transition between the singlet and CT state. The consequence being that although we are monitoring the LE emission, we are actually probing the lifetime of the CT state. We have a section discussing this phenomenon in the main text, and we simulate it in the SI. However, to make the table self-explainable, we have expanded the Table caption. The new caption reads:

"Table 1. Spectroscopic data of the BODIPY oligomers collected in different solvents. The probing wavelength for emission decays was close to the emission maximum, and for transient absorption the maximum of the ground state bleach. Note that as the S_1 and CT states in some cases are connected in a dynamic equilibrium, it is not possible to calculate the radiative rate constant of fluorescence from the yield of fluorescence and the fluorescence lifetime."

Regarding the wavelength of monitoring emission decays. The wavelength is at or very close to the maximum of emission. The exact wavelength is stated in the respective decay graph in the SI.

The text in the manuscript explaining how a long emission lifetime is compatible with a low emission quantum yield is shown below:

"The fluorescence quantum yield of the monomer is greatly reduced in ACN compared to toluene. However, the observed emission lifetime is not reduced to the same extent (Table 1, see Supplementary Figures 27-33 for emission decays). This surprising feature has been previously observed,²⁴ and we will try to rationalize it. When recording the emission lifetime for the monomer, a long-lived transient as well as an instrument response function limited signal is present when monitoring the S_1 but not the CS state emission (Supplementary Figures 27 and 29). The rate of charge separation is thus faster than the time resolution of our instrumentation. The long-lived emission must therefore not be due to the directly excited BODIPY unit. Furthermore, when observing the steady state emission as a function of decreasing temperature (Supplementary Figure 34), we see a reduction of the S_1 emission until a plateau value is reached. Based on these two observations, we suggest that the plateau value represent prompt emission from the S_1 state, and the temperature dependent emission represents endothermic charge recombination from the CS state back to the S_1 state. To strengthen this hypothesis, the system was modelled using rate equations (Supplementary

Section 5). In the modelling, it was assumed that the S_1 and CS states are connected by a microscopic reversibility, implying that the ratio of the rate constants between them being equal to the Boltzmann factor. Using the energy difference between the two states as a fitting parameter, both the steady state and time resolved temperature dependent emission could be qualitatively reproduced (Supplementary Figure 35). Furthermore, the fitted energy difference between the S_1 and CS states was 12 kJ/mol (this corresponds to a 28 nm shift compared to the S_1 energy), which is well in line with the emission data (Fig. 3). It is thus very plausible that the observed long lifetime represents the lifetime of the CS state, illustrating the complex dynamics in these systems.”

One can make a similar observation for the dimer, trimer, and tetramer: In going from toluene to DCM as solvent the emission quantum yields drops, but the lifetime even increases! In these cases, no correction for a CT emission is mentioned in the table caption, and only a single emission maximum is given. Fig. 3 shows CT emission only for the monomer (with a second maximum) and dimer (as a shoulder). Excluding the red tail of the emission from lifetime measurements should give a clearer picture.

The same type of kinetic modeling for the oligomers as for the monomer can be done. However, to not become too technical in the main part of the paper, we have placed all details on the kinetic modelling in the SI part of the paper (Supplementary section 4 deals with the correlation between QY_f and the observed lifetime for the monomer and Supplementary section 7 deals with the QY_f and QY_{ISC} for all compounds). We note that as the lifetime of the CT state in DCM is on the same order of magnitude as the lifetime of the S_1 state in toluene it becomes a little bit confusing when analyzing these values by the traditional correlation between lifetime and fluorescence quantum yield. In order to not confuse the readers, we have added statement that such normal analysis is not applicable due the S_1 and CT states being connected in a dynamic equilibrium.

Page 2, line 10: The sentence breaks off after “regardless”.

We would like to thank the reviewer for spotting this typo, it has been corrected.

Page 7, near bottom: The energy gap law says that the Franck-Condon weighted density of states (FCWD) decays nearly exponential with the energy gap for a given system. I.e., the FCWD for two molecules that have a different number of normal vibrations are by no means equal at the same value of the energy gap. Hence, the energy gap law can not “explain” a decrease of a transition rate by an increased energy gap in comparing two different molecules.

This is true in theory. However, the energy gap law, derived by Jortner, was motivated by the work of Siebrand, who plotted the non-radiative rate constants for a set of differently sized aromatic molecules as a function of the S_0 - S_1 energy. Thus, the experimental validity of the energy gap law comes from molecules having different sizes. We therefore think that we also could use it. In the updated manuscript we have also added a reference to the work of Siebrand (J. Chem. Phys. 47, 2411–2422 (1967)) when we are discussing the energy gap law.

Reviewer #4:

In the present manuscript the authors propose a strategy to shift the singlet excited states of photosensitizers by forming an excitonic J-aggregates while keeping the energy of a triplet state constant. The strategy is demonstrated experimentally on a series of covalently linked bodipy-antracene dyes.

The work is in my opinion suitable for publication in Nature Communications. However, prior to the publication the authors should address the following issues:

1. J-aggregation as a strategy for shifting the singlet state energy without affecting the triplet state is not new and has been reported in the literature as early as in 1939. In the paper by Kautsky and Merkel published in *Naturwissenschaften*, 27, 195 (1939) the authors report that the aggregation of dyes facilitates their action as photophysical sensitizers in photochemical reactions. In fact, this work has motivated later Kasha's work on exciton theory whose initial motivation was to explain enhanced triplet generation in J-aggregates. This old seminal work should be clearly acknowledged.

We thank the author for making us aware of this paper, and has in the updated manuscript commented this work appropriately. The update reads:

“Even before the development of the theory of exciton coupling it was observed that aggregation affects the fluorescence and phosphorescence differently.³³ Curiosity about this observation amongst other things, later lead to the finding that exciton coupling influences the singlet state energy without affecting either the triplet or CS state energies significantly.³⁴”

2. The spectra in Fig 1. clearly show the shift of the excitation energy which is typical for J-aggregates. However, I do not see clearly the exchange narrowing effect that should be also present. The authors might try to quantify the widths of the bright exciton state and compare the results with the prediction of the exciton model in order to see if the trend is correctly reproduced.

This is an insightful suggestion. But it seems that conformational flexibility obscures the narrowing effect in our systems. Although these molecules have methyl groups that prevent planarization, they are not locked in an absolute 90-degree conformation. A distribution of angles around 90 degrees are therefore expected at room temperature, and with that come small contributions from electronic coupling to the S_1 energy. These effects do not seem to be large, but they are enough to broaden the absorbance spectra. We have added a paragraph in the conclusion section where we outline design methods that could improve the photophysical properties of exciton coupled oligomers. Here we suggest the incorporation of spiro-carbons as a means to lower conformational flexibility in order to see the exchange narrowing. The updated paragraph reads:

“Based on this work, the following conclusions on further design concepts to optimize the photophysical performances can be drawn. Electron delocalization is here restricted through steric bulk. This method is easily synthetically implemented but allows for a distribution of dihedral angles between monomers (thus a small but variable amount of electron delocalization), resulting in broader than expected absorption and emission spectral envelopes. By locking the conformation between monomers in an orthogonal orientation, for instance through the introduction of spiro-carbons, electron delocalization is minimized, and a significant narrowing of the absorbance and emission spectra is expected. The benefits of the SOCT-ISC mechanism are that the lifetime of the acquired triplet state is long, and that the method works in absence of heavy atoms. The drawback is that it depends on an intermediate CT state, which energy is highly solvent polarity dependent. Furthermore, in exciton coupled systems, the driving forces for reaching the triplet state is reduced at the same time as the rate of charge recombination to the ground state is increased, lowering the yield of triplet formation. However, the yields of emission are high and lifetimes relatively long for all

oligomers in toluene solution, indicating that exciton coupling itself does not significantly deactivate the excited state. Thus, limitations for the SOCT-ISC process can be overcome by the utilization of other methods of increasing ISC, for instance through the incorporation of heavy atoms.”

2. I am puzzled by the fact that TDDFT calculations based on the long-range corrected functional so strongly overestimate the energy shift in Fig. 1b. This might be due to a rather small basis set used (6-31g(d)). I think that calculations using a triple-zeta basis set should be feasible for this systems and the authors might try to perform them in order to improve the agreement.

We agree with the reviewer that a better match between experiments and theory is expected with increased levels of theory. However, due to the large size of the tetramer molecule (254 atoms) it is for us not computationally feasible to improve on the level of density functional theory substantially, e.g., by increasing basis sets. A single frequency analysis has a CPU runtime of 22 days on a state-of-the-art computational cluster, corresponding to about 17 hours on a 32 core node.

We have nevertheless tried to improve upon our DFT calculations by including an implicit consideration of a CH₂Cl₂ solvent environment via the polarizable continuum model (PCM). A minor (3.5 cm⁻¹) imaginary frequency persist in our model of the tetramer after numerous convergence attempts at this level of theory. This convergence issue is exceedingly small and does not affect our estimates of the adiabatic singlet triplet gaps, which remains constant at 1.49 eV across all molecules. However, since we have this imaginary frequency, we have not included these new results into the manuscript.

3. It would interesting to provide pictures of the electronic transition densities for the bright states.

We do not see how the picture of the electronic transition densities would drive the discussion forward in the manuscript.

4. The authors mention the finding that for J-aggregates reduced reorganization energy counteracts the effect of the energy gap law citing their own work from 2021. This fact has been first theoretically predicted by Scharf and Dinur in 1984 (Chem. Phys. Lett., 105, 78 (1984)). Moreover, the modified energy gap law for J-aggregates has been developed and the size dependence of the radiative and non-radiative rates in J-aggregates has been predicted in J. Phys. Chem. A, 124, 10143 (2020). These works should be properly acknowledged.

These two studies are now referred to in the updated manuscript.

Altogether I find the finding in the present paper sufficiently interesting and novel to warrant the publication in Nat. Comm.

We thank the reviewer for this positive assessment.

Reviewer #1 (Remarks to the Author):

I have reviewed the revised version of the manuscript by Börjesson and co-workers, along with the revised Supporting Information and the authors' response to my initial comments in the rebuttal letter. Despite the authors' efforts to address the concerns raised, I still find that the presented results do not sufficiently justify publication in Nature Communications.

As noted in my initial review, oligomerization of BODIPY appears to have a detrimental effect on their performance as photosensitizers. The triplet state yield values indicate a significant drop when moving from the monomer to the oligomers. Specifically: monomer - 90%; dimer - 24%; trimer and Tetramer - a few percent. Even after attempting to optimize these values by choosing a different solvent, the yields remain unsatisfactory for practical applications. The drastic reduction in triplet state yield for the dimer (24%) compared to the monomer (90%) is particularly concerning. For the trimer and tetramer, the yields are too low to be useful. Therefore, I disagree with the authors' claim that this approach allows for shifting the singlet excited state to lower energies "without compromising the ability of the sensitizer to do work" (a statement made in the Abstract).

In response to my comments, the authors added a new paragraph in the Conclusion section: "Based on this work, the following conclusions on further design concepts to optimize the photophysical performances can be drawn...", discussing design strategies to enhance the photophysical properties of exciton-coupled oligomers. However, these statements are general and have been made in previous works. The current manuscript does not provide data to support these conclusions. For example: "By locking the conformation between monomers in an orthogonal orientation, for instance through the introduction of spirocarbons, electron delocalization is minimized, and a significant narrowing of the absorbance and emission spectra is expected." AND "Thus, limitations for the SOCT-ISC process can be overcome by the utilization of other methods of increasing ISC, for instance through the incorporation of heavy atoms." The manuscript does not implement the introduction of spirocarbons or heavy atoms, and thus these suggestions, while valid, are not substantiated by the current study.

My concern about the presence of impurities in the NMR spectra has not been adequately addressed. The authors stated that size exclusion chromatography (SEC) was used twice for purification, but SEC was not mentioned in the methods section of the main manuscript. Despite these efforts, impurity peaks are still evident in the provided spectra. If SEC did not yield pure samples, alternative purification methods such as HPLC should be considered.

While the manuscript reports some interesting observations regarding the photophysics of BODIPY and the effect of oligomerization on intersystem crossing (ISC), I find the overall significance of these results limited. The substantial decrease in triplet state yields for the oligomers and the lack of demonstration of the applicability of these compounds diminish the impact of the work.

In summary, the manuscript provides valuable insights but does not meet the high standards required for publication in Nature Communications. I suggest the authors consider submitting their work to a specialized journal focused on photophysics/photochemistry, where the detailed investigation of BODIPY oligomers and their photophysical properties might be more suitable.

Reviewer #3 (Remarks to the Author):

This is a revised version of a manuscript that I refereed a month ago. The authors have answered most of my questions and those of the other referees. I think that the paper is written clearly and makes proper reference to previous work in this field. According to these criteria it is acceptable for publication.

The idea of shifting the optically allowed singlet state to lower energies by excitonic coupling has been proposed before, however in a different context. The authors simulate the observed shifts well using experimental transition dipoles and the dipole-dipole coupling formula. In my view this is convincing evidence that the idea worked. This design concept seems, however, to be limited by the requirement of a CS state that should remain lower in energy than the shifted excitonic state.

The explanation of the constantly long lifetime of fluorescence despite strong quenching of the quantum yield is interesting and convincing. In fact, it amounts to the proposal of an E-type delayed fluorescence, however with a singlet state as the reservoir state instead of a triplet state. I am not aware whether such a mechanism has been proposed before. This might be really new.

Whether the novelty of the research is sufficient for the requirements of Nature communications should be the decision of the editor.

Reviewer #4 (Remarks to the Author):

The authors have satisfactorily answered most of the questions that I have raised. I am thus happy to recommend the manuscript for publication in Nature Communications.

Reviewer #1:

I have reviewed the revised version of the manuscript by Börjesson and co-workers, along with the revised Supporting Information and the authors' response to my initial comments in the rebuttal letter. Despite the authors' efforts to address the concerns raised, I still find that the presented results do not sufficiently justify publication in Nature Communications.

As noted in my initial review, oligomerization of BODIPY appears to have a detrimental effect on their performance as photosensitizers. The triplet state yield values indicate a significant drop when moving from the monomer to the oligomers. Specifically: monomer - 90%; dimer - 24%; trimer and Tetramer - a few percent. Even after attempting to optimize these values by choosing a different solvent, the yields remain unsatisfactory for practical applications. The drastic reduction in triplet state yield for the dimer (24%) compared to the monomer (90%) is particularly concerning. For the trimer and tetramer, the yields are too low to be useful. Therefore, I disagree with the authors' claim that this approach allows for shifting the singlet excited state to lower energies "without compromising the ability of the sensitizer to do work" (a statement made in the Abstract).

We thank Reviewer 1 for their comments. We agree that it is disappointing that the triplet quantum yield of the oligomers is much lower than the monomer. However, we still think that the work is valuable and important. It is, as far as we know, the first work that actively attempts to utilize exciton coupling for the purpose of improving the energy efficiency of triplet photosensitizers. We have demonstrated a methodology for evaluating such systems and have shown that we can selectively lower the singlet excited state energy without perturbing the energy of the triplet state. Importantly, note that the sentence cited by the reviewer above refers to the energy of the triplet state and not the yield of forming triplets (it is therefore a valid statement). We hope that this work can now inspire other groups to utilize this design principle and improve the triplet quantum yield in a more suitable system

In response to my comments, the authors added a new paragraph in the Conclusion section: "Based on this work, the following conclusions on further design concepts to optimize the photophysical performances can be drawn...", discussing design strategies to enhance the photophysical properties of exciton-coupled oligomers. However, these statements are general and have been made in previous works. The current manuscript does not provide data to support these conclusions. For example: "By locking the conformation between monomers in an orthogonal orientation, for instance through the introduction of spirocarbons, electron delocalization is minimized, and a significant narrowing of the absorbance and emission spectra is expected." AND "Thus, limitations for the SOCT-ISC process can be overcome by the utilization of other methods of increasing ISC, for instance through the incorporation of heavy atoms." The manuscript does not implement the introduction of spirocarbons or heavy atoms, and thus these suggestions, while valid, are not substantiated by the current study.

We agree that our added discussion on possible future design directions could be improved. In our updated manuscript we have added references to earlier works in which our suggestions are successfully implemented in other contexts. For instance, the use of spirocarbons to lock dyes in a perpendicular position. Furthermore, we have changed our wording to highlight that we are discussing future possibilities in this section.

“We suggest the following design concepts to optimize the photophysical performances beyond those presented herein. Electron delocalization is here restricted through steric bulk. This method is easily synthetically implemented but allows for a distribution of dihedral angles between monomers (thus a small but variable amount of electron delocalization), resulting in broader than expected absorption and emission spectral envelopes. By locking the conformation between monomers in an orthogonal orientation, for instance through the introduction of spiro-carbons, electron delocalization is minimized, and a significant narrowing of the absorbance and emission spectra is expected. Spiro-carbons have previously been introduced in for instance fluorene to form spirobifluorene, which crystal structure shows perpendicular directions of the two fluorene moieties in the dimer.⁵⁷ The benefits of the SOCT-ISC mechanism are that the lifetime of the acquired triplet state is long, and that the method works in absence of heavy atoms. The drawback is that it depends on an intermediate CT state, which energy is highly solvent polarity dependent.¹⁸ Furthermore, in exciton coupled systems, the driving force for reaching the triplet state is reduced at the same time as the rate of charge recombination to the ground state is increased, lowering the yield of triplet formation. However, the yields of emission are high and lifetimes relatively long for all oligomers in toluene solution, indicating that exciton coupling itself does not significantly deactivate the excited state. Thus, limitations for the SOCT-ISC process can most likely be overcome by the utilization of other methods of increasing ISC, for instance through the incorporation of heavy atoms.”

My concern about the presence of impurities in the NMR spectra has not been adequately addressed. The authors stated that size exclusion chromatography (SEC) was used twice for purification, but SEC was not mentioned in the methods section of the main manuscript. Despite these efforts, impurity peaks are still evident in the provided spectra. If SEC did not yield pure samples, alternative purification methods such as HPLC should be considered.

We have now done analytical HPLC on all our molecules in order to have an assessment of the purity of the molecules. The Table below shows the purity of the molecules, and the respective chromatograms are shown at the end of this document. The information on the purity is incorporated in the caption of the NMR spectra in the SI. In short, the trimer is of high purity, and the tetramer is less pure. We do not know if it is contaminations or atropoisomers that we are seeing in the chromatogram of the tetramer. As the purity is affecting the measurement of the molar absorptivity, we have in the updated manuscript written a note that the lower-than-expected measured transition dipole moment (Fig 2c) probably is due to the purity of the sample. This note reads:

“As expected for an exciton coupled system, the transition dipole moments scale well with the square root of the number of moieties (Fig. 2c),³⁵ although limited purity of the tetramer (Supplementary Figure 17) is exhibited as a less than expected transition dipole moment for that molecule.”

However, we do not believe that other measurements are significantly affected, this because we are both taking transient spectra and decays, so we know which species we are measuring on when recording our decays. Furthermore, the tetramer streaked very much on the analytical column used and we have so far not been able to clean the column completely. Because we do believe that our photophysical measurements are not significantly affected by the lower purity, we would like to avoid injecting this compound on a preparatory column. This since we do these purifications on another groups instrument that we do not want to contaminate.

Table showing measured purities by analytical HPLC.

Molecule	Purity
Monomer	98%
Dimer	97%
Trimer_1	95%
Trimer_2	98%
Tetramer	84%

While the manuscript reports some interesting observations regarding the photophysics of BODIPY and the effect of oligomerization on intersystem crossing (ISC), I find the overall significance of these results limited. The substantial decrease in triplet state yields for the oligomers and the lack of demonstration of the applicability of these compounds diminish the impact of the work. In summary, the manuscript provides valuable insights but does not meet the high standards required for publication in Nature Communications. I suggest the authors consider submitting their work to a specialized journal focused on photophysics/photochemistry, where the detailed investigation of BODIPY oligomers and their photophysical properties might be more suitable.

We kindly disagree with the reviewer's assessment of the impact of our work. We believe our work to be conceptually novel and of importance and interest to the scientific community.

Reviewer #3:

This is a revised version of a manuscript that I refereed a month ago. The authors have answered most of my questions and those of the other referees. I think that the paper is written clearly and makes proper reference to previous work in this field. According to these criteria it is acceptable for publication.

The idea of shifting the optically allowed singlet state to lower energies by excitonic coupling has been proposed before, however in a different context. The authors simulate the observed shifts well using experimental transition dipoles and the dipole-dipole coupling formula. In my view this is convincing evidence that the idea worked. This design concept seems, however, to be limited by the requirement of a CS state that should remain lower in energy than the shifted excitonic state.

The explanation of the constantly long lifetime of fluorescence despite strong quenching of the quantum yield is interesting and convincing. In fact, it amounts to the proposal of an E-type delayed fluorescence, however with a singlet state as the reservoir state instead of a triplet state. I am not aware whether such a mechanism has been proposed before. This might be really new.

Whether the novelty of the research is sufficient for the requirements of Nature communications should be the decision of the editor.

We would like to thank Reviewer 3 for assessing our manuscript. We agree with the reviewer that our proposed mechanism that describes the reason for the observed long fluorescence lifetime of the monomer is similar to that of E-type delayed fluorescence. Furthermore, we are like the reviewer not aware of any previous work discussing such a mechanism, and are pleased to read that the reviewer are of the opinion that these discussions enhances the impact of our work. In the updated manuscript we now added a note to highlight the strong similarity between our system and that of E-type delayed fluorescence (modelling wise). The added sentence reads:

“The modelled system thus strongly resembles that used when modelling molecules exhibiting E-type delayed fluorescence (with the triplet state then being replaced by a CS state).”

Reviewer #4:

The authors have satisfactorily answered most of the questions that I have raised. I am thus happy to recommend the manuscript for publication in Nature Communications.

We would like to thank Reviewer 4 for assessing our manuscript.

Figure R1. HPLC chromatogram of the Monomer sample.

Figure R2. HPLC chromatogram of the Dimer sample.

Figure R3. HPLC chromatogram of the Trimer1 sample.

Figure R4. HPLC chromatogram of the Trimer2 sample.

Figure R5. HPLC chromatogram of the Tetramer sample.